# Learning Optimal Commitment to Overcome Insecurity

**Avrim Blum**
Carnegie Mellon University
avrim@cs.cmu.edu

**Nika Haghtalab**
Carnegie Mellon University
nika@cmu.edu

**Ariel D. Procaccia**
Carnegie Mellon University
arielpro@cs.cmu.edu

## Abstract

Game-theoretic algorithms for physical security have made an impressive real-world impact. These algorithms compute an optimal strategy for the defender to commit to in a Stackelberg game, where the attacker observes the defender's strategy and best-responds. In order to build the game model, though, the payoffs of potential attackers for various outcomes must be estimated; inaccurate estimates can lead to significant inefficiencies. We design an algorithm that optimizes the defender's strategy with no prior information, by observing the attacker's responses to randomized deployments of resources and learning his priorities. In contrast to previous work, our algorithm requires a number of queries that is polynomial in the representation of the game.

## 1 Introduction

The US Coast Guard, the Federal Air Marshal Service, the Los Angeles Airport Police, and other major security agencies are currently using game-theoretic algorithms, developed in the last decade, to deploy their resources on a regular basis [13]. This is perhaps the biggest practical success story of computational game theory — and it is based on a very simple idea. The interaction between the *defender* and a potential *attacker* can be modeled as a *Stackelberg game*, in which the defender commits to a (possibly randomized) deployment of his resources, and the attacker responds in a way that maximizes his own payoff. The algorithmic challenge is to compute an optimal defender strategy — one that would maximize the defender's payoff under the attacker's best response.

While the foregoing model is elegant, implementing it requires a significant amount of information. Perhaps the most troubling assumption is that we can determine the attacker's payoffs for different outcomes. In deployed applications, these payoffs are estimated using expert analysis and historical data — but an inaccurate estimate can lead to significant inefficiencies. The uncertainty about the attacker's payoffs can be encoded into the optimization problem itself, either through robust optimization techniques [12], or by representing payoffs as continuous distributions [5].

Letchford et al. [8] take a different, learning-theoretic approach to dealing with uncertain attacker payoffs. Studying Stackelberg games more broadly (which are played by two players, a *leader* and a *follower*), they show that the leader can *efficiently learn* the follower's payoffs by iteratively committing to different strategies, and observing the attacker's sequence of responses. In the context of security games, this approach may be questionable when the attacker is a terrorist, but it is a perfectly reasonable way to calibrate the defender's strategy for routine security operations when the attacker is, say, a smuggler. And the learning-theoretic approach has two major advantages over modifying the defender's optimization problem. First, the learning-theoretic approach requires no prior information. Second, the optimization-based approach deals with uncertainty by inevitably degrading the quality of the solution, as, intuitively, the algorithm has to simultaneously optimize against a range of possible attackers; this problem is circumvented by the learning-theoretic approach.

But let us revisit what we mean by "efficiently learn". The number of queries (i.e., observations of follower responses to leader strategies) required by the algorithm of Letchford et al. [8] is polynomial in the number of pure leader strategies. The main difficulty in applying their results to Stackelberg

security games is that even in the simplest security game, the number of pure defender strategies is exponential in the representation of the game. For example, if each of the defender's resources can protect one of two potential targets, there is an exponential number of ways in which resources can be assigned to targets. [1]

**Our approach and results.** We design an algorithm that learns an (additively) $\epsilon$-optimal strategy for the defender with probability $1 - \delta$, by asking a number of queries that is polynomial in the representation of the security game, and logarithmic in $1/\epsilon$ and $1/\delta$. Our algorithm is completely different from that of Letchford et al. [8]. Its novel ingredients include:

- We work in the space of feasible coverage probability vectors, i.e., we directly reason about the probability that each potential target is protected under a randomized defender strategy. Denoting the number of targets by $n$, this is an $n$-dimensional space. In contrast, Letchford et al. [8] study the exponential-dimensional space of randomized defender strategies. We observe that, in the space of feasible coverage probability vectors, the region associated with a specific best response for the attacker (i.e., a specific target being attacked) is convex.

- To optimize within each of these convex regions, we leverage techniques — developed by Tauman Kalai and Vempala [14] — for optimizing a linear objective function in an unknown convex region using only membership queries. In our setting, it is straightforward to build a membership oracle, but it is quite nontrivial to satisfy a key assumption of the foregoing result: that the optimization process starts from an interior point of the convex region. We do this by constructing a hierarchy of nested convex regions, and using smaller regions to obtain interior points in larger regions.

- We develop a method for efficiently discovering new regions. In contrast, Letchford et al. [8] find regions (in the high-dimensional space of randomized defender strategies) by sampling uniformly at random; their approach is inefficient when some regions are small.

## 2  Preliminaries

A Stackelberg security game is a two-player general-sum game between a *defender* (or the *leader*) and an *attacker* (or the *follower*). In this game, the defender commits to a randomized allocation of his security resources to defend potential targets. The attacker, in turn, observes this randomized allocation and attacks the target with the best expected payoff. The defender and the attacker receive payoffs that depend on the target that was attacked and whether or not it was defended. The defender's goal is to choose an allocation that leads to the best payoff.

More precisely, a security game is defined by a 5-tuple $(T, \mathcal{D}, R, A, U)$:

- $T = \{1, \ldots, n\}$ is a set of $n$ *targets*.
- $R$ is a set of *resources*.
- $\mathcal{D} \subseteq 2^T$ is a collection of subsets of targets, each called a *schedule*, such that for every schedule $D \in \mathcal{D}$, targets in $D$ can be simultaneously defended by one resource. It is natural to assume that if a resource is capable of covering schedule $D$, then it can also cover any subset of $D$. We call this property *closure under the subset operation*; it is also known as "subsets of schedules are schedules (SSAS)" [7].
- $A : R \to 2^{\mathcal{D}}$, called the *assignment function*, takes a resource as input and returns the set of all schedules that the resource is capable of defending. An allocation of resources is *valid* if every resource $r$ is allocated to a schedule in $A(r)$.
- The *payoffs* of the players are given by functions $U_d(t, p_t)$ and $U_a(t, p_t)$, which return the expected payoffs of the defender and the attacker, respectively, when target $t$ is attacked and it is covered with probability $p_t$ (as formally explained below). We make two assumptions that are common to all papers on security games. First, these utility functions are linear. Second, the attacker prefers it if the attacked target is not covered, and the defender prefers

it if the attacked target is covered, i.e., $U_d(t, p_t)$ and $U_a(t, p_t)$ are respectively increasing and decreasing in $p_t$. We also assume w.l.o.g. that the utilities are normalized to have values in $[-1, 1]$. If the utility functions have coefficients that are rational with denominator at most $a$, then the game's (utility) representation length is $L = n \log n + n \log a$.

A *pure strategy* of the defender is a valid assignment of resources to schedules. The set of pure strategies is determined by $T$, $\mathcal{D}$, $R$, and $A$. Let there be $m$ pure strategies; we use the following $n \times m$, zero-one matrix $M$ to represent the set of all pure strategies. Every row in $M$ represents a target and every column represents a pure strategy. $M_{ti} = 1$ if and only if target $t$ is covered using some resource in the $i^{th}$ pure strategy. A *mixed strategy* (hereinafter, called strategy) is a distribution over the pure strategies. To represent a strategy we use a $1 \times m$ vector $\mathbf{s}$, such that $s_i$ is the probability with which the $i^{th}$ strategy is played, and $\sum_{i=1}^{m} s_i = 1$.

Given a defender's strategy, the *coverage probability* of a target is the probability with which it is defended. Let $\mathbf{s}$ be a defender's strategy, then the coverage probability vector is $\mathbf{p}^T = M\mathbf{s}^T$, where $p_t$ is coverage probability of target $t$. We call a probability vector *implementable* if there exists a strategy that imposes that coverage probability on the targets.

Let $\mathbf{p}^s$ be the corresponding coverage probability vector of strategy $\mathbf{s}$. The attacker's *best response* to $\mathbf{s}$ is defined by $b(\mathbf{s}) = \arg\max_t U_a(t, p_t^s)$. Since the attacker's best-response is determined by the coverage probability vector irrespective of the strategy, we slightly abuse notation by using $b(\mathbf{p}^s)$ to denote the best-response, as well. We say that target $t$ is "better" than $t'$ for the defender if the highest payoff he receives when $t$ is attacked is more than the highest payoff he receives when $t'$ is attacked. We assume that if multiple targets are tied for the best-response, then ties are broken in favor of the "best" target.

The defender's *optimal strategy* is defined as the strategy with highest expected payoff for the defender, i.e. $\arg\max_\mathbf{s} U_d(b(\mathbf{s}), p_{b(\mathbf{s})}^\mathbf{s})$. An optimal strategy $\mathbf{p}$ is called *conservative* if no other optimal strategy has a strictly lower sum of coverage probabilities. For two coverage probability vectors we use $\mathbf{q} \preceq \mathbf{p}$ to denote that for all $t$, $q_t \leq p_t$.

## 3 Problem Formulation and Technical Approach

In this section, we give an overview of our approach for learning the defender's optimal strategy when $U_a$ is not known. To do so, we first review how the optimal strategy is computed in the case where $U_a$ is known.

Computing the defender's optimal strategy, even when $U_a(\cdot)$ is known, is NP-Hard [6]. In practice the optimal strategy is computed using two formulations: Mixed Integer programming [11] and Multiple Linear Programs [1]; the latter provides some insight for our approach. The Multiple LP approach creates a separate LP for every $t \in T$. This LP, as shown below, solves for the optimal defender strategy under the restriction that the strategy is valid (second and third constraints) and the attacker best-responds by attacking $t$ (first constraint). Among these solutions, the optimal strategy is the one where the defender has the highest payoff.

$$
\begin{aligned}
\text{maximize} \quad & U_d\big(t, \sum_{i:M_{ti}=1} s_i\big) \\
\text{s.t.} \quad & \forall t' \neq t, \; U_a\big(t', \sum_{i:M_{t'i}=1} s_i\big) \leq U_a\big(t, \sum_{i:M_{ti}=1} s_i\big) \\
& \forall i, \; s_i \geq 0 \\
& \sum_{i=1}^{n} s_i = 1
\end{aligned}
$$

We make two changes to the above LP in preparation for finding the optimal strategy in polynomially many queries, when $U_a$ is unknown. First, notice that when $U_a$ is unknown, we do not have an *explicit* definition of the first constraint. However, *implicitly* we can determine whether $t$ has a better payoff than $t'$ by observing the attacker's best-response to $\mathbf{s}$. Second, the above LP has exponentially

many variables, one for each pure strategy. However, given the coverage probabilities, the attacker's actions are independent of the strategy that induces that coverage probability. So, we can restate the LP to use variables that represent the coverage probabilities and add a constraint that enforces the coverage probabilities to be implementable.

$$
\begin{aligned}
\text{maximize} \quad & U_d(t, p_t) \\
\text{s.t.} \quad & t \text{ is attacked} \\
& \mathbf{p} \text{ is implementable}
\end{aligned}
\tag{1}
$$

This formulation requires optimizing a linear function over a region of the space of coverage probabilities, by using membership queries. We do so by examining some of the characteristics of the above formulation and then leveraging an algorithm introduced by Tauman Kalai and Vempala [14] that optimizes over a convex set, using only an initial point and a membership oracle. Here, we restate their result in a slightly different form.

**Theorem 2.1** [14, restated]. *For any convex set $H \subseteq \mathbb{R}^n$ that is contained in a ball of radius $R$, given a membership oracle, an initial point with margin $r$ in $H$, and a linear function $\ell(\cdot)$, with probability $1 - \delta$ we can find an $\epsilon$-approximate optimal solution for $\ell$ in $H$, using $O(n^{4.5} \log \frac{nR^2}{r\epsilon\delta})$ queries to the oracle.*

## 4 Main Result

In this section, we design and analyze an algorithm that $(\epsilon, \delta)$-learns the defender's optimal strategy in a number of best-response queries that is polynomial in the number of targets and the representation, and logarithmic in $\frac{1}{\epsilon}$ and $\frac{1}{\delta}$. Our main result is:

**Theorem 1.** *Consider a security game with $n$ targets and representation length $L$, such that for every target, the set of implementable coverage probability vectors that induce an attack on that target, if non-empty, contains a ball of radius $1/2^L$. For any $\epsilon, \delta > 0$, with probability $1 - \delta$, Algorithm 2 finds a defender strategy that is optimal up to an additive term of $\epsilon$, using $O(n^{6.5}(\log \frac{n}{\epsilon\delta} + L))$ best-response queries to the attacker.*

The main assumption in Theorem 1 is that the set of implementable coverage probabilities for which a given target is attacked is either empty or contains a ball of radius $1/2^L$. This implies that if it is possible to make the attacker prefer a target, then it is possible to do so with a small margin. This assumption is very mild in nature and its variations have appeared in many well-known algorithms. For example, interior point methods for linear optimization require an initial feasible solution that is within the region of optimization with a small margin [4]. Letchford et al. [8] make a similar assumption, but their result depends linearly, instead of logarithmically, on the minimum volume of a region (because they use uniformly random sampling to discover regions).

To informally see why such an assumption is necessary, consider a security game with $n$ targets, such that an attack on any target but target 1 is very harmful to the defender. The defender's goal is therefore to convince the attacker to attack target 1. The attacker, however, only attacks target 1 under a very specific coverage probability vector, i.e., the defender's randomized strategy has to be just so. In this case, the defender's optimal strategy is impossible to approximate.

The remainder of this section is devoted to proving Theorem 1. We divide our intermediate results into sections based on the aspect of the problem that they address. The proofs of most lemmas are relegated to the appendix; here we mainly aim to provide the structure of the theorem's overall proof.

### 4.1 Characteristics of the Optimization Region

One of the requirements of Theorem 2.1 is that the optimization region is convex. Let $\mathcal{P}$ denote the space of implementable probability vectors, and let $\mathcal{P}_t = \{\mathbf{p} : \mathbf{p} \text{ is implementable and } b(\mathbf{p}) = t\}$. The next lemma shows that $\mathcal{P}_t$ is indeed convex.

**Lemma 1.** *For all $t \in T$, $\mathcal{P}_t$ is the intersection of a finitely many half-spaces.*

*Proof.* $\mathcal{P}_t$ is defined by the set of all $\mathbf{p} \in [0,1]^n$ such that there is $\mathbf{s}$ that satisfies the LP with the following constraints. There are $m$ half-spaces of the form $s_i \geq 0$, 2 half-spaces $\sum_i s_i \leq 1$ and

$\sum_i s_i \geq 1$, $2n$ half-spaces of the form $M\mathbf{s}^T - \mathbf{p}^T \leq 0$ and $M\mathbf{s}^T - \mathbf{p}^T \geq 0$, and $n-1$ half-spaces of the form $U_a(t, p_t) - U_a(t', p_{t'}) \geq 0$. Therefore, the set of $(\mathbf{s}, \mathbf{p}) \in \mathcal{R}^{m+n}$ such that $\mathbf{p}$ is implemented by strategy $\mathbf{s}$ and causes an attack on $t$ is the intersection of $3n+m+1$ half-spaces. $\mathcal{P}_t$ is the reflection of this set on $n$ dimensions; therefore, it is also the intersection of at most $3n+m+1$ half-spaces. $\qquad\square$

Lemma 1, in particular, implies that $\mathcal{P}_t$ is convex. The Lemma's proof also suggests a method for finding the minimal half-space representation of $\mathcal{P}$. Indeed, the set $S = \{(\mathbf{s}, \mathbf{p}) \in \mathbb{R}^{m+n} :$ Valid strategy $\mathbf{s}$ implements $\mathbf{p}\}$ is given by its half-space representation. Using the Double Description Method [2, 10], we can compute the vertex representation of $S$. Since, $\mathcal{P}$ is a linear transformation of $S$, its vertex representation is the transformation of the vertex representation of $S$. Using the Double Description Method again, we can find the minimal half-space representation of $\mathcal{P}$.

Next, we establish some properties of $\mathcal{P}$ and the half-spaces that define it. The proofs of the following two lemmas appear in Appendices A.1 and A.2, respectively.

**Lemma 2.** *Let $\mathbf{p} \in \mathcal{P}$. Then for any $\mathbf{0} \preceq \mathbf{q} \preceq \mathbf{p}$, $\mathbf{q} \in \mathcal{P}$.*

**Lemma 3.** *Let $A$ be a set of a positive volume that is the intersection of finitely many half-spaces. Then the following two statements are equivalent.*

1. *For all $\mathbf{p} \in A$, $\mathbf{p} \succeq \boldsymbol{\epsilon}$. And for all $\boldsymbol{\epsilon} \preceq \mathbf{q} \preceq \mathbf{p}$, $\mathbf{q} \in A$.*

2. *$A$ can be defined as the intersection of $\mathbf{e}_i \cdot \mathbf{p} \geq \epsilon$ for all $i$, and a set $H$ of half-spaces, such that for any $\mathbf{h} \cdot \mathbf{p} \geq b$ in $H$, $\mathbf{h} \preceq \mathbf{0}$, and $b \leq -\epsilon$.*

Using Lemmas 2 and 3, we can refer to the set of half-spaces that define $\mathcal{P}$ by $\{(\mathbf{e}_i, 0) :$ for all $i\} \cup H_{\mathcal{P}}$, where for all $(\mathbf{h}^*, b^*) \in H_{\mathcal{P}}$, $\mathbf{h}^* \preceq \mathbf{0}$, and $b^* \leq 0$.

## 4.2 Finding Initial Points

An important requirement for many optimization algorithms, including the one developed by Tauman Kalai and Vempala [14], is having a "well-centered" initial feasible point in the region of optimization. There are two challenges involved in discovering an initial feasible point in the interior of every region. First, establishing that a region is non-empty, possibly by finding a boundary point. Second, obtaining a point that has a significant margin from the boundary. We carry out these tasks by executing the optimization in a hierarchy of sets where at each level the optimization task only considers a subset of the targets and the feasibility space. We then show that optimization in one level of this hierarchy helps us find initial points in new regions that are well-centered in higher levels of the hierarchy.

To this end, let us define *restricted regions*. These regions are obtained by first perturbing the defining half-spaces of $\mathcal{P}$ so that they conform to a given representation length, and then trimming the boundaries by a given width (See Figure 1).

In the remainder of this paper, we use $\gamma = \frac{1}{(n+1)2^{L+1}}$ to denote the accuracy of the representation and the width of the trimming procedure for obtaining restricted regions. More precisely:

**Definition 1** (restricted regions). *The set $\mathcal{R}^k \in \mathbb{R}^n$ is defined by the intersection the following half-spaces: For all $i$, $(\mathbf{e}_i, k\gamma)$. For all $(\mathbf{h}^*, b^*) \in H_{\mathcal{P}}$, a half-space $(\mathbf{h}, b + k\gamma)$, such that $\mathbf{h} = \gamma\lfloor \frac{1}{\gamma}\mathbf{h}^* \rfloor$ and $b = \gamma\lceil \frac{1}{\gamma}b^* \rceil$. Furthermore, for every $t \in T$, define $\mathcal{R}_t^k = \mathcal{R}^k \cap \mathcal{P}_t$.*

The next Lemma, whose proof appears in Appendix A.3, shows that the restricted regions are subsets of the feasibility space, so, we can make best-response queries within them.

**Lemma 4.** *For any $k \geq 0$, $\mathcal{R}^k \subseteq \mathcal{P}$.*

The next two lemmas, whose proofs are relegated to Appendices A.4 and A.5, show that in $\mathcal{R}^k$ one can reduce each coverage probability individually down to $k\gamma$, and the optimal conservative strategy in $\mathcal{R}^k$ indeed reduces the coverage probabilities of all targets outside the best-response set to $k\gamma$.

**Lemma 5.** *Let $\mathbf{p} \in \mathcal{R}^k$, and let $\mathbf{q}$ such that $k\boldsymbol{\gamma} \preceq \mathbf{q} \preceq \mathbf{p}$. Then $\mathbf{q} \in \mathcal{R}^k$.*

**Lemma 6.** *Let $\mathbf{s}$ and its corresponding coverage probability $\mathbf{p}$ be a conservative optimal strategy in $\mathcal{R}^k$. Let $t^* = b(\mathbf{s})$ and $B = \{t : U_a(t, p_t) = U_a(t^*, p_{t^*})\}$. Then for any $t \notin B$, $p_t = k\gamma$.*

| Target | Attacker | Defender |
|--------|----------|----------|
| 1 | $0.5(1 - p_1)$ | $-0.5(1 - p_1)$ |
| 2 | $(1 - p_2)$ | $-(1 - p_2)$ |

(a) Utilities of the game

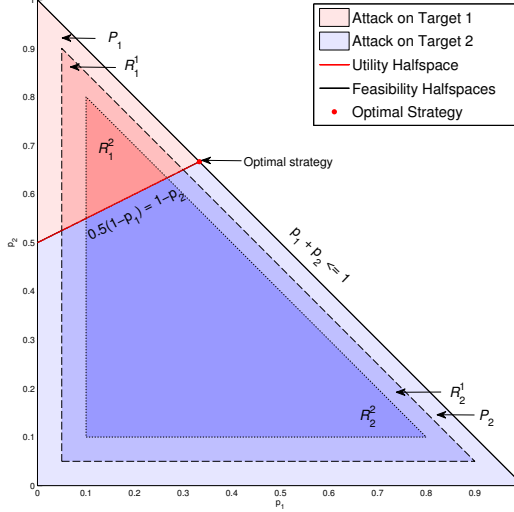

(b) Regions

Figure 1: A security game with one resource that can cover one of two targets. The attacker receives utility 0.5 from attacking target 1 and utility 1 from attacking target 2, when they are not defended; he receives 0 utility from attacking a target that is being defended. The defender's utility is the zero-sum complement.

The following Lemma, whose proof appears in Appendix A.6 shows that if every non-empty $\mathcal{P}_t$ contains a large enough ball, then $\mathcal{R}_t^n \neq \emptyset$.

**Lemma 7.** *For any $t$ and $k \leq n$ such that $\mathcal{P}_t$ contains a ball of radius $r > \frac{1}{2^L}$, $\mathcal{R}_t^k \neq \emptyset$.*

The next lemma provides the main insight behind our search for the region with the highest-paying optimal strategy. It implies that we can restrict our search to strategies that are optimal for a subset of targets in $\mathcal{R}^{\bar{k}}$, if the attacker also agrees to play within that subset of targets. At any point, if the attacker chooses a target outside the known regions, he is providing us with a point in a new region. Crucially, Lemma 8 requires that we optimize exactly inside each restricted region, and we show below (Algorithm 1 and Lemma 11) that this is indeed possible.

**Lemma 8.** *Assume that for every $t$, if $\mathcal{P}_t$ is non-empty, then it contains a ball of radius $\frac{1}{2^L}$. Given $K \subseteq T$ and $k \leq n$, let $\mathbf{p} \in \mathcal{R}^k$ be the coverage probability of the strategy that has $k\gamma$ probability mass on targets in $T \setminus K$ and is optimal if the attacker were to be restricted to attacking targets in $K$. Let $\mathbf{p}^*$ be the optimal strategy in $\mathcal{P}$. If $b(\mathbf{p}) \in K$ then $b(\mathbf{p}^*) \in K$.*

*Proof.* Assume on the contrary that $b(\mathbf{p}^*) = t^* \notin K$. Since $\mathcal{P}_{t^*} \neq \emptyset$, by Lemma 7, there exists $\mathbf{p}' \in \mathcal{R}_{t^*}^k$.

For ease of exposition, replace $\mathbf{p}$ with its corresponding conservative strategy in $\mathcal{R}^k$. Let $B$ be the set of targets that are tied for the attacker's best-response in $\mathbf{p}$, i.e. $B = \arg\max_{t \in T} U_a(t, p_t)$. Since $b(\mathbf{p}) \in K$ and ties are broken in favor of the "best" target, i.e. $t^*$, it must be that $t^* \notin B$. Then, for any $t \in B$, $U_a(t, p_t) > U_a(t^*, k\gamma) \geq U_a(t^*, p'_{t^*}) \geq U_a(t, p'_t)$. Since $U_a$ is decreasing in the coverage probability, for all $t \in B$, $p'_t > p_t$. Note that there is a positive gap between the attacker's payoff for attacking a best-response target versus another target, i.e. $\Delta = \min_{t' \in K \setminus B, t \in B} U_a(t, p_t) - U_a(t', p_{t'}) > 0$, so it is possible to increase $p_t$ by a small amount without changing the best response. More precisely, since $U_a$ is continuous and decreasing in the coverage probability, for every $t \in B$, there exists $\delta < p'_t - p_t$ such that for all $t' \in K \setminus B$, $U_a(t', p_{t'}) < U_a(t, p'_t - \delta) < U_a(t, p_t)$.

Let $\mathbf{q}$ be such that for $t \in B$, $q_t = p'_t - \delta$ and for $t \notin B$, $q_t = p_t = k\gamma$ (by Lemma 6 and the fact that $\mathbf{p}$ was replaced by its conservative equivalent). By Lemma 5, $\mathbf{q} \in \mathcal{R}^k$. Since for all $t \in B$ and $t' \in K \setminus B$, $U_a(t, q_t) > U_a(t', q_{t'})$, $b(\mathbf{q}) \in B$. Moreover, because $U_d$ is increasing in the coverage probability for all $t \in B$, $U_d(t, q_t) > U_d(t, p_t)$. So, $\mathbf{q}$ has higher payoff for the defender when the attacker is restricted to attacking $K$. This contradicts the optimality of $\mathbf{p}$ in $\mathcal{R}^k$. Therefore, $b(\mathbf{p}^*) \in K$. ∎

If the attacker attacks a target $t$ outside the set of targets $K$ whose regions we have already discovered, we can use the new feasible point in $\mathcal{R}_t^k$ to obtain a well-centered point in $\mathcal{R}_t^{k-1}$, as the next lemma formally states.

**Lemma 9.** *For any $k$ and $t$, let $\mathbf{p}$ be any strategy in $\mathcal{R}_t^k$. Define $\mathbf{q}$ such that $q_t = p_t - \frac{\gamma}{2}$ and for all $i \neq t$, $q_i = p_i + \frac{\gamma}{4\sqrt{n}}$. Then, $\mathbf{q} \in \mathcal{R}_t^{k-1}$ and $\mathbf{q}$ has distance $\frac{\gamma}{2n}$ from the boundaries of $\mathcal{R}_t^{k-1}$.*

The lemma's proof is relegated to Appendix A.7.

## 4.3 An Oracle for the Convex Region

We use a three-step procedure for defining a membership oracle for $\mathcal{P}$ or $\mathcal{R}_t^k$. Given a vector $\mathbf{p}$, we first use the half-space representation of $\mathcal{P}$ (or $\mathcal{R}^k$) described in Section 4.1 to determine whether $\mathbf{p} \in \mathcal{P}$ (or $\mathbf{p} \in \mathcal{R}^k$). We then find a strategy $\mathbf{s}$ that implements $\mathbf{p}$ by solving a linear system with constraints $M\mathbf{s}^T = \mathbf{p}^T$, $\mathbf{0} \preceq \mathbf{s}$, and $\|\mathbf{s}\|_1 = 1$. Lastly, we make a best-response query to the attacker for strategy $\mathbf{s}$. If the attacker responds by attacking $t$, then $\mathbf{p} \in \mathcal{P}_t$ (or $\mathbf{p} \in \mathcal{R}_t^k$), else $\mathbf{p} \notin \mathcal{P}_t$ (or $\mathbf{p} \notin \mathcal{R}_t^k$).

## 4.4 The Algorithms

In this section, we define algorithms that use the results from previous sections to prove Theorem 1. First, we define Algorithm 1, which receives an approximately optimal strategy in $\mathcal{R}_t^k$ as input, and finds the optimal strategy in $\mathcal{R}_t^k$. As noted above, obtaining exact optimal solutions in $\mathcal{R}_t^k$ is required in order to apply Lemma 8, thereby ensuring that we discover new regions when lucrative undiscovered regions still exist.

---

**Algorithm 1** LATTICE-ROUNDING (approximately optimal strategy $\mathbf{p}$)

1. For all $i \neq t$, make best-response queries to binary search for the smallest $p_i' \in [k\gamma, p_i]$ up to accuracy $\frac{1}{2^{5n(L+1)}}$, such that $t = b(\mathbf{p}')$, where for all $j \neq i$, $p_j' \leftarrow p_j$.

2. For all $i$, set $r_i$ and $q_i$ respectively to the smallest and second smallest rational numbers with denominator at most $2^{2n(L+1)}$, that are larger than $p_i' - \frac{1}{2^{5n(L+1)}}$.

3. Define $\mathbf{p}^*$ such that $p_t^*$ is the unique rational number with denominator at most $2^{2n(L+1)}$ in $[p_t, p_t + \frac{1}{2^{4n(L+1)}})$. (Refer to the proof for uniqueness), and for all $i \neq t$, $p_i^* \leftarrow r_i$.

4. Query $j \leftarrow b(\mathbf{p}^*)$.

5. If $j \neq t$, let $p_j^* \leftarrow q_i$. Go to step 4

6. Return $\mathbf{p}^*$.

---

The next two Lemmas, whose proofs appear in Appendices A.8 and A.9, establish the guarantees of Algorithm 1. The first is a variation of a well-known result in linear programming [3] that is adapted specifically for our problem setting.

**Lemma 10.** *Let $\mathbf{p}^*$ be a basic optimal strategy in $\mathcal{R}_t^k$, then for all $i$, $p_i^*$ is a rational number with denominator at most $2^{2n(L+1)}$.*

**Lemma 11.** *For any $k$ and $t$, let $\mathbf{p}$ be a $\frac{1}{2^{6n(L+1)}}$-approximate optimal strategy in $\mathcal{R}_t^k$. Algorithm 1 finds the optimal strategy in $\mathcal{R}_t^k$ in $O(nL)$ best-response queries.*

At last, we are ready to prove our main result, which provides guarantees for Algorithm 2, given below.

**Theorem 1** (restated). *Consider a security game with $n$ targets and representation length $L$, such that for every target, the set of implementable coverage probability vectors that induce an attack on that target, if non-empty, contains a ball of radius $1/2^L$. For any $\epsilon, \delta > 0$, with probability $1 - \delta$, Algorithm 2 finds a defender strategy that is optimal up to an additive term of $\epsilon$, using $O(n^{6.5}(\log \frac{n}{\epsilon\delta} + L))$ best-response queries to the attacker.*

*Proof Sketch.* For each $K \subseteq T$ and $k$, the loop at step 5 of Algorithm 2 finds the optimal strategy if the attacker was restricted to attacking targets of $K$ in $\mathcal{R}^k$.

Every time the IF clause at step 5a is satisfied, the algorithm expands the set $K$ by a target $t'$ and adds $\mathbf{x}^{t'}$ to the set of initial points $X$, which is an interior point of $\mathcal{R}_{t'}^{k-1}$ (by Lemma 9). Then the algorithm restarts the loop at step 5. Therefore every time the loop at step 5 is started, $X$ is a set of initial points in $K$ that have margin $\frac{\gamma}{2n}$ in $\mathcal{R}^k$. This loop is restarted at most $n - 1$ times.

We reach step 6 only when the best-response to the optimal strategy that only considers targets of $K$ is in $K$. By Lemma 8, the optimal strategy is in $\mathcal{P}_t$ for some $t \in K$. By applying Theorem 2.1 to $K$,

---

**Algorithm 2** OPTIMIZE (accuracy $\epsilon$, confidence $\delta$)

1. $\gamma \leftarrow \frac{1}{(n+1)2^{L+1}}$, $\delta' \leftarrow \frac{\delta}{n^2}$, and $k \leftarrow n$.

2. Use $R, \mathcal{D}$, and $A$ to compute oracles (half-spaces) for $\mathcal{P}, \mathcal{R}^0, \ldots, \mathcal{R}^n$.

3. Query $t \leftarrow b(k\boldsymbol{\gamma})$

4. $K \leftarrow \{t\}$, $X \leftarrow \{\mathbf{x}^t\}$, where $x_t^t = k\gamma - \gamma/2$ and for $i \neq t$, $x_i^t = k\gamma + \frac{\gamma}{4\sqrt{n}}$.

5. For $t \in K$,

   (a) If during steps 5b to 5e a target $t' \notin K$ is attacked as a response to some strategy $\mathbf{p}$:

      i. Let $x_{t'}^{t'} \leftarrow p_{t'} - \gamma/2$ and for $i \neq t'$, $x_i^{t'} \leftarrow p_i + \frac{\gamma}{4\sqrt{n}}$.

      ii. $X \leftarrow X \cup \{\mathbf{x}^{t'}\}$, $K \leftarrow K \cup \{t'\}$, and $k \leftarrow k - 1$.

      iii. Restart the loop at step 5.

   (b) Use Theorem 2.1 with set of targets $K$. With probability $1 - \delta'$ find a $\mathbf{q}^t$ that is a $\frac{1}{2^{6n(L+1)}}$-approximate optimal strategy restricted to set $K$.

   (c) Use the Lattice Rounding on $\mathbf{q}^t$ to find $\mathbf{q}^{t*}$, that is the optimal strategy in $\mathcal{R}_t^k$ restricted to $K$.

   (d) For all $t' \notin K$, $q_{t'}^{t*} \leftarrow k\gamma$.

   (e) Query $\mathbf{q}^{t*}$.

6. For all $t \in K$, use Theorem 2.1 to find $\mathbf{p}^{t*}$ that is an $\epsilon$-approximate strategy with probability $1 - \delta'$, in $\mathcal{P}_t$.

7. Return $\mathbf{p}^{t*}$ that has the highest payoff to the defender.

---

with an oracle for $\mathcal{P}$ using the initial set of point $X$ which has $\gamma/2n$ margin in $\mathcal{R}^0$, we can find the $\epsilon$-optimal strategy with probability $1 - \delta'$. There are at most $n^2$ applications of Theorem 2.1 and each succeeds with probability $1 - \delta'$, so our overall procedure succeeds with probability $1 - n^2\delta' \geq 1 - \delta$.

Regarding the number of queries, every time the loop at step 5 is restarted $|K|$ increases by 1. So, this loop is restarted at most $n - 1$ times. In a successful run of the loop for set $K$, the loop makes $|K|$ calls to the algorithm of Theorem 2.1 to find a $\frac{1}{2^{6n(L+1)}}$-approximate optimal solution. In each call, $X$ has initial points with margin $\frac{\gamma}{2n}$, and furthermore, the total feasibility space is bounded by a sphere of radius $\sqrt{n}$ (because of probability vectors), so each call makes $O(n^{4.5}(\log \frac{n}{\delta} + L))$ queries. The last call looks for an $\epsilon$-approximate solution, and will take another $O(n^{4.5}(\log \frac{n}{\epsilon\delta} + L))$ queries. In addition, our the algorithm makes $n^2$ calls to Algorithm 1 for a total of $O(n^3 L)$ queries. In conclusion, our procedure makes a total of $O(n^{6.5}(\log \frac{n}{\epsilon\delta} + L)) = \text{poly}(n, L, \log \frac{1}{\epsilon\delta})$ queries. $\quad\square$

## 5  Discussion

Our main result focuses on the query complexity of our problem. We believe that, indeed, best response queries are our most scarce resource, and it is therefore encouraging that an (almost) optimal strategy can be learned with a polynomial number of queries.

It is worth noting, though, that some steps in our algorithm are computationally inefficient. Specifically, our membership oracle needs to determine whether a given coverage probability vector is implementable. We also need to explicitly compute the feasibility half-spaces that define $\mathcal{P}$. Informally speaking, (worst-case) computational inefficiency is inevitable, because computing an optimal strategy to commit to is computationally hard even in simple security games [6].

Nevertheless, deployed security games algorithms build on integer programming techniques to achieve satisfactory runtime performance in practice [13]. While beyond the reach of theoretical analysis, a synthesis of these techniques with ours can yield truly practical learning algorithms for dealing with payoff uncertainty in security games.

**Acknowledgments.** This material is based upon work supported by the National Science Foundation under grants CCF-1116892, CCF-1101215, CCF-1215883, and IIS-1350598.

## Footnotes

[1]Subsequent work by Marecki et al. [9] focuses on exploiting revealed information during the learning process — via Monte Carlo Tree Search — to optimize total leader payoff. While their method provably converges to the optimal leader strategy, no theoretical bounds on the rate of convergence are known.

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
