[Supplementary Material]

# Appendix

## A   Omitted Proofs of Lemmas

### A.1   Proof of Lemma 2

Let there be $k$ targets $t$ such that $p_t > q_t$. We prove this lemma by induction on $k$. For $k = 0$, the lemma trivially holds. Assume that for all $k < k_0$ the result holds. Let $k = k_0$. Let $t$ be an arbitrary target for which $p_t > q_t$. Since the set of pure strategies is closed under subset operation there is a map, $\sigma(\cdot)$, such that for every pure strategy $M_i$ (a column in matrix $M$) such that $M_{ti} = 1$, $M_{\sigma(i)}$ only differs from $M_i$ in the $t^{th}$ row (target). Let $I$ and $I'$ indicate these strategies i.e. $I = \{i : M_{ti} = 1\}$, $I' = \{\sigma(i) : M_{ti} = 1\}$.

Define $\mathbf{s}'$ as follows: For all $i \in I$, let $s'_i = s_i \cdot \frac{q_t}{p_t}$ and $s'_{\sigma(i)} = s_{\sigma(i)} + s_i \cdot \frac{p_t - q_t}{q_t}$, and for any $i \notin I \cup I'$, let $s'_i = s_i$. Consider $\mathbf{p}'$ that is induced by $\mathbf{s}'$: For $t$, $p'_t = \sum_{i:M_{ti}=1} s'_i = \sum_{i \in I} s_i \cdot \frac{q_t}{p_t} = q_t$. For all $t' \neq t$,

$$
\begin{aligned}
p'_{t'} &= \sum_{i:M_{t'i}=1} s'_i = \sum_{\substack{i:M_{t'i}=1 \\ \text{and } i \in I}} \left(s'_i + s'_{\sigma(i)}\right) + \sum_{\substack{i:M_{t'i}=1 \\ \text{and } i \notin I \cup I'}} s'_i \\
&= \sum_{\substack{i:M_{t'i}=1 \\ \text{and } i \in I}} \left(\frac{q_t}{p_t} s_i + s_{\sigma(i)} + \frac{p - q_t}{p_t} s_i\right) + \sum_{\substack{i:M_{t'i}=1 \\ \text{and } i \notin I \cup I'}} s_i \\
&= \sum_{\substack{i:M_{t'i}=1 \\ \text{and } i \in I}} \left(s_i + s_{\sigma(i)}\right) + \sum_{\substack{i:M_{t'i}=1 \\ \text{and } i \notin I \cup I'}} s_i = \sum_{i:M_{t'i}=1} s_i \\
&= p_{t'}
\end{aligned}
$$

We conclude that $\mathbf{p}'$ such that $p'_t = q_t$ and for all $t' \neq t$, $p'_{t'} = p_t$, is implementable. $\mathbf{p}'$ and $\mathbf{q}$ differ in only $k_0 - 1$ indices and for all $i$, $p'_i \geq q_i$, so using the induction hypothesis $\mathbf{q}$ is implementable. $\quad\square$

### A.2   Proof of Lemma 3

($1 \implies 2$). Consider the minimal set of half-spaces that defines $A$. We know that this set is unique, and is the collection of facet-defining half-spaces. Since $A$ has a positive volume, for all $i$, $(\mathbf{e}_i, \epsilon)$ is a facet, so it belongs to the set of half-spaces that define $A$. Take any half-space $(\mathbf{h}, b)$ in this collection that is not of the form $(\mathbf{e}_i, \epsilon)$. There is a point $\mathbf{p}$ on the the boundary of $(\mathbf{h}, b)$ that is not on the boundary of any other half-space (including $(\mathbf{e}_i, \epsilon)$), so $\mathbf{p} \succ \boldsymbol{\epsilon}$. For every $i$, define $\mathbf{p}^i$ such that $p_i^i = \epsilon$ and $p_j^i = p_j$ for all $j \neq i$. Then,

$$\mathbf{h} \cdot \mathbf{p}^i = \mathbf{h} \cdot \mathbf{p} + \mathbf{h} \cdot (\mathbf{p}_i^i - \mathbf{p}) = b - h_i(p_i - \epsilon).$$

Since $\mathbf{p}^i \in A$, $h_i \leq 0$. Since, $\boldsymbol{\epsilon} \in A$, $\mathbf{h} \cdot \boldsymbol{\epsilon} = -\|\mathbf{h}\|_1 \epsilon \geq b$, so $b \leq -\epsilon$.

($2 \implies 1$). For any $\mathbf{p} \in A$ and any $\boldsymbol{\epsilon} \preceq \mathbf{q} \preceq \mathbf{p}$, and any $i$, $\mathbf{e}_i \cdot \mathbf{q} = q_i \geq \epsilon$. For any $(h, b) \in H$ of the second form,

$$\mathbf{h} \cdot \mathbf{q} = \mathbf{h} \cdot \mathbf{p} + \mathbf{h} \cdot (\mathbf{q} - \mathbf{p}) \geq b,$$

where the last transition is by the fact that $\mathbf{h}$ and $(\mathbf{q} - \mathbf{p})$ are non-positive. So, $\mathbf{q} \in A$. $\quad\square$

### A.3   Proof of Lemma 4

Let $\mathbf{p} \notin \mathcal{P}$, by Lemma 3 one of the following cases holds: (1) There exists $i$, such that $\mathbf{e}_i \cdot \mathbf{p} < 0$. In this case, $\mathbf{e}_i \cdot \mathbf{p} \leq k\gamma$. So, $\mathbf{p} \notin R^k$. (2) There exists $(\mathbf{h}^*, b^*) \in H_{\mathcal{P}}$ such that $\mathbf{h}^* \cdot \mathbf{p} < b^*$. Let $(\mathbf{h}, b)$ be the corresponding half-space of $(\mathbf{h}^*, b^*)$ in $\mathcal{R}^k$. Then,

$$\mathbf{h} \cdot \mathbf{p} = \mathbf{h}^* \cdot \mathbf{p} + (\mathbf{h} - \mathbf{h}^*) \cdot \mathbf{p} < b^* + (\mathbf{h} - \mathbf{h}^*) \cdot \mathbf{p} < b$$

where the last transition is by the fact that $\mathbf{h} \preceq \mathbf{h}^*$, $b > b^*$, and $\mathbf{p} \succeq \mathbf{0}$. $\quad\square$

## A.4  Proof of Lemma 5

If $\mathcal{R}^k = \emptyset$ is empty then the result holds trivially. If $\mathcal{R}^k \neq \emptyset$, then there exists $\mathbf{p} \succeq k\gamma$ such that $\mathbf{p} \in \mathcal{R}^k$. By Lemmas 2 and 3, $\mathcal{P}$ has half-spaces of the form $(\mathbf{e}_i, 0)$ and $(\mathbf{h}^*, b^*)$, such that $\mathbf{h}^* \preceq \mathbf{0}$ and $b^* \leq 0$. By construction of $\mathcal{R}^k$, we have half-spaces of the form $(\mathbf{e}_i, k\gamma)$ for all $i$, and half-spaces $(\mathbf{h}, b + k\gamma)$ such that $\mathbf{h} = \lfloor \frac{1}{\gamma}\mathbf{h}^* \rfloor \preceq \mathbf{h}^*$. Since, $k\gamma \preceq \mathbf{p} \in \mathcal{R}^k$, $\mathbf{h} \cdot k\gamma = \|h\|_1 k\gamma \geq b$. So, $b \leq -k\gamma$. The proof is completed by the fact that the conditions of Lemma 3 hold. $\qquad\square$

## A.5  Proof of Lemma 6

Note that there is a positive gap between the attacker's payoff from attacking a best-response target versus another target, i.e. $\min_{t \notin B} U_a(t^*, p_{t^*}) - U_a(t, p_t) > 0$. Since $U_a$ is continuous and decreasing in the coverage probability, for any $t \notin B$, if $p_t > k\gamma$ there exists $0 < \delta \leq p_t - k\gamma$ such that $U_a(t^*, p_{t^*}) > U_a(t, p_t - \delta)$. Let $\mathbf{q}$ be defined such that for all $t \notin B$, $q_t = p_t - \delta \geq k\gamma$, and for all $t \in B$, $q_t = p_t$. Then by Lemma 5, $\mathbf{q}$ is implementable by some strategy $\mathbf{s}_q$ in $\mathcal{R}^k$. Furthermore, $b(\mathbf{q}) = t^*$, so, $\mathbf{s}_q$ is also an optimal strategy. This contradicts the fact that $\mathbf{s}$ is a conservative optimal strategy. $\qquad\square$

## A.6  Proof of Lemma 7

Let $\mathbf{p}$ be the center of the ball. Then for any $i$, $p_i \geq r \geq k\gamma$, so $\mathbf{e}_i \cdot \mathbf{p} \geq k\gamma$. For any $(\mathbf{h}, b)$ defining half-space of $\mathcal{R}^k$ and its corresponding half-space $(\mathbf{h}^*, b^*)$ of $\mathcal{P}$ we have:

$$\mathbf{h} \cdot \mathbf{p} - b = \mathbf{h}^*\mathbf{p} - b^* + (\mathbf{h} - \mathbf{h}^*) \cdot \mathbf{p} + (b^* - b) \geq r - \gamma n - (\gamma + k\gamma) \geq 0,$$

where the second transition is by the fact that $\mathbf{h}^*\mathbf{p} = r$ is the margin of $\mathbf{p}$ from a normalized half-space $\mathbf{h}^*$, and that for any value $x$ and $1/\gamma \in \mathbb{Z}$, $\gamma\lfloor \frac{1}{\gamma}x \rfloor$ and $\gamma\lceil \frac{1}{\gamma}x \rceil$ are within $\gamma$ from $x$. Hence, $\mathbf{p} \in \mathcal{R}_t^k$. $\qquad\square$

## A.7  Proof of Lemma 9

The boundaries of $\mathcal{R}^k$ are defined by $(\mathbf{e}_i, k\gamma)$ for all $i$ and a half-spaces $(\mathbf{h}, b + k\gamma)$ for every half-space $(\mathbf{h}^*, b^*) \in H_{\mathcal{P}}$ such that $\mathbf{h} = \gamma\lfloor \frac{1}{\gamma}\mathbf{h}^* \rfloor$ and $b = \gamma\lceil \frac{1}{\gamma}b^* \rceil$. In addition $\mathcal{R}_t^k$ is the intersection of $\mathcal{R}^k$ with half-spaces $U_t(t, p_t) \geq U(t', p_{t'})$ for all $t' \neq t$. Let $\mathrm{dist}(\cdot)$ denote the signed distance of a point from a half-space. For every $i \neq t$,

$$\mathrm{dist}(\mathbf{q}, (\mathbf{e}_i, (k-1)\gamma)) = \frac{\mathbf{e}_i \cdot \mathbf{q} - (k-1)\gamma}{\|\mathbf{e}_i\|_2} = q_i - (k-1)\gamma \geq p_i + \frac{\gamma}{4\sqrt{n}} - (k-1)\gamma$$

$$\geq k\gamma + \frac{\gamma}{4\sqrt{n}} - (k-1)\gamma > \frac{\gamma}{2n}.$$

Moreover,

$$\mathrm{dist}(\mathbf{q}, (\mathbf{e}_t, (k-1)\gamma)) = \frac{\mathbf{e}_t \cdot \mathbf{q} - (k-1)\gamma}{\|\mathbf{e}_t\|_2} = q_t - (k-1)\gamma = p_t - \frac{\gamma}{2} - (k-1)\gamma$$

$$\geq k\gamma - \frac{\gamma}{2} - (k-1)\gamma \geq \frac{\gamma}{2n}.$$

Finally, for every $(\mathbf{h}, b + (k-1)\gamma)$,

$$\mathrm{dist}(\mathbf{q}, (\mathbf{h}, b + (k-1)\gamma)) = \frac{\mathbf{h} \cdot \mathbf{q} - (b + (k-1)\gamma)}{\|\mathbf{h}\|_2} \geq \frac{\mathbf{h} \cdot \mathbf{p} + h \cdot (\mathbf{q} - \mathbf{p}) - (b + (k-1)\gamma)}{2}$$

$$\geq \frac{h \cdot (\mathbf{q} - \mathbf{p}) + \gamma}{2} \geq -\frac{\gamma}{8\sqrt{n}}\|\mathbf{h}\|_1 - h_t\left(\frac{\gamma}{4} + \frac{\gamma}{8\sqrt{n}}\right) + \frac{\gamma}{2}$$

$$\geq -\frac{\gamma}{8\sqrt{n}}\|\mathbf{h}\|_1 + \frac{\gamma}{2} \geq -\frac{\gamma}{8\sqrt{n}}(\sqrt{n} + n\gamma) + \frac{\gamma}{2}$$

$$\geq \frac{3\gamma}{8} - \frac{\sqrt{n}\gamma^2}{8} \geq \frac{\gamma}{2n},$$

where the first inequality is by the fact that $\|\mathbf{h}\|_2 \leq \|\mathbf{h}^*\|_2 + \gamma\sqrt{n} \leq 1 + \gamma\sqrt{n} < 2$ (by the triangle inequality), the penultimate inequality is by the fact that $\|\mathbf{h}\|_1 \leq \|\mathbf{h}^*\|_1 + n\gamma \leq \sqrt{n}\|\mathbf{h}^*\|_2 + n\gamma$, and the last inequality follows from $\gamma = \frac{1}{(n+1)2^{L+1}} < \frac{1}{n\sqrt{n}}$.

As for the utility half-spaces of the form $U_a(t, q_t) - U_a(t', q_{t'}) \geq 0$, for every $t'$, the probability $\mathbf{q}$ has moved away by at least $\min(\frac{\gamma}{2}, \frac{\gamma}{4\sqrt{n}}) \geq \frac{\gamma}{2n}$ from every half-space. Moreover, by reducing the coverage probability on the attacked target and increasing it on other targets, the attacker receives even larger payoff from attacking $t$, so $\mathbf{q}$ still induces an attack on $t$, i.e. $\mathbf{q} \in \mathcal{P}_t$. Finally, the signed distance of $\mathbf{q}$ from every half-space is greater than $\frac{\gamma}{2n}$, therefore $\mathbf{q} \in \mathcal{R}^{k-1} \cap \mathcal{P}_t = \mathcal{R}_t^{k-1}$, and it has distance $\frac{\gamma}{2n}$ from the boundaries of $\mathcal{R}_t^{k-1}$. □

## A.8 Proof of Lemma 10

Let $\mathcal{R}_t^k$ be represented as a system $\{\mathbf{p} : A\mathbf{p}^T \succeq \mathbf{b}\}$ where there is a row (constraint) for each half-space that defines $\mathcal{R}^k$, and a row for each $t' \neq t$ of the form $U_a(t, p_t) - U_a(t', p_{t'}) \geq 0$. Furthermore, assume that $A$ is normalized so that every row has integral coefficients. Note that by the definition of the game representation length, each coefficient of the utility rows is at most $2^L$. Moreover, by the definition of the defining half-spaces of $\mathcal{R}^k$, each coefficients of the feasibility constraints are at most $(n+1)2^{L+1}$.

We know that each basic solution to the above LP is at the intersection of $n$ independent constraints of $A$. Let $\mathbf{p}^*$ be such a solution. Let $D$ represent those $n$ hyper-planes. Using Cramer's rule, for all $i$, $p_i^* = \frac{\det(D_i)}{\det(D)}$, where the $D_i$ is $D$ with its $i^{th}$ column replaced by $b$. Using Hadamard's inequality,

$$\det(D) \leq \prod_{i=1}^n \sqrt{\sum_{j=1}^n d_{ij}^2} \leq \prod_{i=1}^n (n+1)2^{L+1}\sqrt{n} \leq n^{2n}2^{n(L+1)} \leq 2^{2n(L+1)},$$

Where the last inequality is by the fact that $L > n\log n$. □

## A.9 Proof of Lemma 11

Let $\mathbf{p}^*$ be the optimal strategy in $\mathcal{R}_t^k$. By Lemma 10, for all $i$, $p_i^*$ has a denominator of at most $2^{2n(L+1)}$. Note that the difference between two distinct rational numbers with denominators at most $2^{2n(L+1)}$ is at least $\frac{1}{2^{4n(L+1)}}$.

Strategy $\mathbf{p}$ is a $\frac{1}{2^{6n(L+1)}}$-approximate optimal strategy and the utilities have representation length of at most $L$, so

$$p_t^* - p_t \leq 2^L \cdot \frac{1}{2^{6n(L+1)}} < \frac{1}{2^{4n(L+1)}}.$$

Therefore, $p_t^* \in [p_t, p_t + \frac{1}{2^{4n(L+1)}})$. Since this range is smaller than the difference between two rational numbers with denominator at most $2^{2n(L+1)}$, there is at most one such rational number in this range, to which our algorithms sets $p_t^*$. Note that the absence of such a rational number is contradictory to Lemma 10 or the fact that $\mathbf{p}$ was a $\frac{1}{2^{6n(L+1)}}$-approximate optimal strategy.

For all $i$, let $p_i' \geq k\gamma$ be the smallest coverage probability, with accuracy $\frac{1}{2^{5n(L+1)}}$, such that $U_a(t, p_t) \geq U_a(i, p_i')$. Then, $p_i^* \geq p_i' - \frac{1}{2^{5n(L+1)}}$. Let $r_i$ and $q_i$, respectively, be the smallest and second smallest rational numbers with denominator at most $2^{2n(L+1)}$ in the range $[p_i' - \frac{1}{2^{5n(L+1)}}, 1)$. We claim that $p_i^* = r_i$ or $q_i$. To prove this claim, it is sufficient to show that for all $i$, $U_a(i, q_i) \leq U_a(t, p_t^*)$. Since $q_i$ is the second smallest rational number with denominator at most $2^{2n(L+1)}$ in the given range, then $q_i \geq p_i' - \frac{1}{2^{5n(L+1)}} + \frac{1}{2^{4n(L+1)}} > \frac{1}{2^{4n(L+1)+1}}$. Then,

$$U_a(i, p_i') - U_a(i, q_i) \geq \frac{1}{2^L}(q_i - p_i') \geq \frac{1}{2^{4n(L+1)+L+1}}$$

$$> \frac{2^L}{2^{6n(L+1)}} \geq 2^L \left(U_d(t, p_t^*) - U_d(t, p_t)\right)$$

$$> U_a(t, p_t) - U_a(t, p_t^*).$$

Since, $U_a(t, p_t) \geq U_a(i, p'_i)$, the above inequality implies that $U_a(i, q_i) \leq U_a(t, p^*_t)$. So, for each $i$, it is sufficient to query the attacker to see whether $U_a(i, r_i) \leq U_a(t, p^*_t)$ if so then $p^*_i = r_i$, else $p^*_i = q_i$.

For each $i$, this algorithm makes $O(\log 2^{5(L+1)}) = O(L)$ queries to find $p'_i$ with accuracy $\frac{1}{2^{5n(L+1)}}$. Values of $p^*_t$, $r_i$ and $q_i$ are computed without any best-response queries. Because $p^*_i = r_i$ or $q_i$, step 4, is repeated at most $n$ times, so there are $n$ additional queries. In conclusion, our algorithm makes $O(nL + n) = O(nL)$ many queries in total. $\qquad\qquad\square$