[Reviews · NeurIPS 2014]

Submitted by Assigned_Reviewer_5

Motivated by the practical problem of designing a security deployment strategy to protect targets from an adversary the author(s) model and study this as a Stackelberg game. The main result of the author(s) is that the defender can efficiently learn the payoffs of the adversary by carefully deploying resources and observing the adversary's attacks. Clearly, this setting may not be viable in the cases where the cost incurred by the defender on a successful attack is large (such as a terrorist attack) but perhaps is a reasonable strategy for other cases such as drug smuggling. The main result of the paper is a "probably approximately optimal" algorithm that finds a defender optimal strategy by learning from polynomial (in the number of targets and encoding length of the problem) number of "attacks" from the adversary. The previous results require the number of observations to grow polynomially in the number of protection strategies which the authors show can be exponential in the number of targets to protect. The proof is a delicate reduction to an algorithm of Kalai-Vempala to optimize over an unknown convex set using only membership queries.

The paper presents an original advance over existing techniques and is very well written considering the technical details.

Minor comments:
* I did not find the introduction of the terms "leader" & "follower" to help the reader at all since it is used interchangeably with defender & attacker. If the main use is to connect your work with the Stackelberg game perhaps a footnote would suffice?
* There's a forward reference to covering probability in the definition in the payoff function part of Section 2.
Summary: This paper presents an important theoretical advance for security games and I recommend the paper for acceptance.

Submitted by Assigned_Reviewer_31

This paper considers the problem of learning an attacker’s payoffs in a security game. In deployed application, game-theoretic solutions to security games have had much success but crucially rely upon experts to provide the payoffs for the attackers (terrorists, drug smugglers, etc) to the equilibrium solvers. This paper suggests replacing this expert-provided knowledge by learning the payoffs from the attacker by observing attacker responses in repeated plays of the attacker-defender game. The authors make use of techniques by Kalai and Vempala which uses an initial feasible point and a membership oracle to optimize the defender’s objective. To do so, the authors prove in this paper that the optimization region is reasonably well-behaved, most notably that there exists initial points that are sufficiently far from the boundaries of the optimization region. Using all of this, they show that the optimal defender strategy can be computed with a polynomial number of queries (plays of the security game), improving over the previous result of an exponential number of plays.

On the high level, I think this paper has a good idea of attempting to correct for the uncertainty in payoff knowledge and uses sophisticated techniques in their chosen approach. If the best way to correct for this uncertainty was to explore the strategy space and find the exact optimal solution, then this paper does offer a significant improvement over the prior method to do so. Indeed, the techniques in this paper is non-trivial and has some nice applications of convex optimization methods to game theoretic settings, i.e. the interpretation of the attacker’s best response as the membership oracle.

However, I’m not convinced that this approach is the right way of solving this problem and the authors don’t make an argument for it. Its not clear that a polynomial number of plays of this security game represents a reasonable amount of computation in the studied contexts. What is the rate that such games occur? Do attacker utility functions remain constant for long enough to actually learn in this context?

I think this work could have benefited from empirical analysis in a couple of ways. First, I would have been very interested to see the practical convergence rate on some typical instances of these games (for example, instances of the games used for the empirical testing in [1]). Perhaps the convergence rate is practically very fast, so the concern about the number of games played is not necessary. Second, it would have been great to see the proposed method compared to some other strategies used to address this uncertainty like the robust optimization methods or distribution knowledge methods as mentioned by the authors in the introduction. Maybe its the case that the method in this paper finds such a better solution that its worthwhile to wait for the convergence of this algorithm.

Finally, for realistic scenarios, is finding the exact optimal solution is the right objective, especially if the defender sacrifices a lot of interim utility in order to find it? A concept like regret might be better suited as it allows for the examination of the trade-offs between exploring the entire strategy-space versus exploiting good strategies that may be learned very quickly.

[1]: Jain, Manish, et al. "Software assistants for randomized patrol planning for the lax airport police and the federal air marshal service." Interfaces 40.4 (2010): 267-290.
Summary: This paper addresses a good problem within the realm of security games and is a nice application of convex optimization techniques to security games. However, it isn't convincing that the algorithm in this paper is actually a good way to solve this problem.

Submitted by Assigned_Reviewer_37

This paper provides a solid theoretical contribution to learn the optimal strategy in security games approximately. The proposed algorithm and analysis are based on a technique by Kalai and Vempala and a non-trivial membership oracle that is designed exclusively for security games, which gives us a surprising exponential speedup compared to a previous work [8]. The paper is well written and the proofs appear to be correct. I believe that the introduction of this new technique will stimulate further research on learning security games.

P2 Tauman Kalai -> Kalai
Summary: The paper is well written and the proofs appear to be correct. I believe that the introduction of this new technique will stimulate further research on learning security games.
Author Feedback
Author rebuttal: We thank all the reviewers for the helpful reviews.

RESPONSE TO REVIEWER 1 (masked ID 31):

The reviewer wonders whether learning an optimal strategy in a polynomial number of rounds "represents a reasonable amount of computation in the studied contexts". We agree that in some relevant settings, this would not be reasonable. But as we briefly argue in the introduction, we are motivated by routine security tasks, where the attacker is, e.g., a drug dealer or a smuggler. When the loss from an attack is not large, a polynomial "calibration phase" does seem reasonable. As the reviewer notes, another implicit assumption is that the attacker's utility function stays fixed. While specific attackers may change over time, we are thinking of a setting where the attacker "type" stays fixed (e.g., we are dealing with smugglers). The attacker's utility function is also affected by the infrastructure of the defended site, which plausibly stays fixed. Interactions satisfying all these properties arise frequently in the context of several of the organizations that use game-theoretic security algorithms, such as the US Coast Guard and the LAX Airport Police. We plan to flesh out this argument in the paper's introduction.

As for regret minimization, note that while our algorithm's number of queries is bounded by a polynomial in the game's parameters, it is only logarithmic in 1/epsilon. Therefore, setting epsilon=1/T gives, by definition, a no-regret algorithm with a superb (in theory) convergence rate of O((log T)/T). In other words, when T is large, it is natural to think of the learning phase as the "exploration phase", which is followed by an "exploitation stage" in which the epsilon-optimal strategy is played repeatedly. That said, we agree that it is natural to ask whether there exist anytime algorithms that achieve low regret even when T is small, by interleaving exploration and exploitation.